# Biodeterioration of Untreated Polypropylene Microplastic Particles by Antarctic Bacteria

**DOI:** 10.3390/polym12112616

**Published:** 2020-11-06

**Authors:** Syahir Habib, Anastasia Iruthayam, Mohd Yunus Abd Shukor, Siti Aisyah Alias, Jerzy Smykla, Nur Adeela Yasid

**Affiliations:** 1Department of Biochemistry, Faculty of Biotechnology and Biomolecular Sciences, Universiti Putra Malaysia, Serdang 43400, Selangor, Malaysia; syahirhabib@gmail.com (S.H.); anastasia1219@outlook.my (A.I.); mohdyunus@upm.edu.my (M.Y.A.S.); 2Institute of Ocean and Earth Sciences, C308 Institute of Postgraduate Studies, University of Malaya, Kuala Lumpur 50603, Malaysia; saa@um.edu.my; 3National Antarctic Research Centre, B303 Level 3, Block B, IPS Building, Universiti Malaya, Kuala Lumpur 50603, Malaysia; 4Institute of Nature Conservation, Polish Academy of Sciences, Mickiewicza, 33, 31-120 Kraków, Poland; jerzysmykla@yahoo.com

**Keywords:** plastic pollution, pristine fellfield soil, polypropylene, *Pseudomonas* sp., *Rhodococcus* sp.

## Abstract

Microplastic pollution is globally recognised as a serious environmental threat due to its ubiquitous presence related primarily to improper dumping of plastic wastes. While most studies have focused on microplastic contamination in the marine ecosystem, microplastic pollution in the soil environment is generally little understood and often overlooked. The presence of microplastics affects the soil ecosystem by disrupting the soil fertility and quality, degrading the food web, and subsequently influencing both food security and human health. This study evaluates the growth and biodegradation potential of the Antarctic soil bacteria *Pseudomonas* sp. ADL15 and *Rhodococcus* sp. ADL36 on the polypropylene (PP) microplastics in Bushnell Haas (BH) medium for 40 days. The degradation was monitored based on the weight loss of PP microplastics, removal rate constant per day (*K*), and their half-life. The validity of the PP microplastics’ biodegradation was assessed through structural changes via Fourier transform infrared spectroscopy analyses. The weight loss percentage of the PP microplastics by ADL15 and ADL36 after 40 days was 17.3% and 7.3%, respectively. The optimal growth in the BH media infused with PP microplastics was on the 40th and 30th day for ADL15 and ADL36, respectively. The infrared spectroscopic analysis revealed significant changes in the PP microplastics’ functional groups following the incubation with Antarctic strains.

## 1. Introduction

Microplastics are considered as minute plastic particles with a size of <5 mm which cause severe environmental pollution due to their exponential increase in abundance in large-scale manufacturing and mass production and their vast utilisation in the world [1]. Several environmental and health concerns ought to be issued as the sorption feature of microplastic may enhance the toxicity of neighbouring co-pollutants. The presence of these synthetic wastes can also suffer biomagnification in the biota food chain within the successive trophic levels [2,3].

While the pollution of microplastics and the associated environmental aftermath have been extensively studied in marine ecosystems [4,5], the growth of research regarding their adverse effects in terrestrial ecosystems, particularly soil biomes, has only commenced in the past few years [6,7]. The substantial gap of studies between these two biomes is largely due to the distinction in plastic accumulation, the complexity of soil medium, and the intricate procedure of extracting the plastic debris from soil samples [2]. In brief, microplastics can be introduced to the soil environment by the improper waste disposal (later degraded through light and/or weathering), discharge of domestic sewage (microplastic beads from personal care products, microfibers from the domestic washing machine), wastewater treatment effluent, and the practice of plastic mulching [8,9]. When these synthetic wastes enter the soil medium, they initially settle on the surface before penetrating the subsoils [10]. The disintegrated microplastics can then further penetrate deeper soil layers by the act of the soil occupants such as collembolans, earthworms, and plants [7,11,12].

Weathering of microplastics is the major process that determines the polymer degradation rate and process in the ecosystem. However, the combined processes of photo-oxidation, soil pH, and surrounding temperature only deteriorate these plastic wastes to a small degree and occur at a very slow yet gradual pace [13,14]. The limited exposure of photo-oxidation towards the soil-incorporated microplastics compared to those on the surface only aggravates the situation. Regarding this matter, microbial biodegradation seems to be most likely method of accelerating the deterioration processes. Certain microorganisms that are found in the deeper soil colonise the plastic polymers, where biodegradation is initiated through the adherence of microorganisms to the plastic surfaces—facilitated via the formation of biofilm [15]. Once colonised, these microbes secrete proteinases to catalyse the hydrolysis of the plastic polymers, resulting in the successive formation of oligomers, monomers, and metabolic intermediates that are ultimately mineralised into carbon dioxide and water molecules [14,16]. However, when discussing the biodegradation of plastics in the soil environment, some factors must be considered. To begin with, the type of soil varies from one location to another. The characteristics of these soils are strictly associated with the uncontrolled biotic and abiotic factors which are highly correlated with the geographical aspects. Due to this, the microbial biodegradation processes can vary from one environment to another and are highly dependent on different climates [14].

Antarctica is generally perceived as one of the last pristine environments on Earth. Nonetheless, Antarctica’s environments are not immune to the anthropogenic impacts occurring both globally and locally within the region. Over the last 3 years, reports on microplastic detection across the maritime Antarctic have shown a dramatic surge [17,18]. While neither clear nor major data have been reported on microplastic contamination in the Antarctic soil, the recent discovery of microplastic particles inside the gut of the Antarctic soil collembolan *Cryptopygus antarcticus* reveals the extent and possibilities of microplastic contamination in the Antarctic terrestrial system [19]. 

Polypropylene (PP) is widely utilised in applications from single-use disposables to long-lasting durables due to its excellent features—exceptional mechanical properties, simple manufacturing, and reasonable price. The improvements to the transparency, strength, and shelf-life in PP production have significantly increased its usage [20]. The continuous chain of repetitive methylene (-CH_2_-) units in PP creates a highly hydrophobic polymer that is extremely resistant to degradation. The addition of the repetitive presence of methyl side chains (-CH_3_) in every unit also decreases the potential of PP to undergo degradation processes [21]. This structural feature of PP hinders the attachment of microbes on its surface, deters the production of biofilm (or biosurfactant), and decelerates the onset of degradation [22,23]. Due to this, several studies have been conducted in an attempt to accelerate the biodegradation rate of PP using supplements such as carbon-high starch [24] and bioplastic blends [25,26]. Efforts also have been made to increase/decrease the hydrophobicity/hydrophilicity of PP by treating the polyolefins to generate more polar groups via thermal treatment, UV exposure, and addition of pro-oxidants [21,27,28,29]. To date, only a single report has presented the degradation of untreated PP by the action of microbes [20]. 

Microorganisms that can grow under extreme temperature and are capable of using neat microplastic as their sole energy source have not been reported yet. Therefore, in this study, we attempt to investigate the ability of two Antarctic hydrocarbon-degrading strains to use polypropylene (PP) microplastics as their sole carbon source. The utilisation of PP by the strains was determined via the growth of the Antarctic bacteria, while the biodegradation of PP was measured via the weight loss and reduction rate after the incubation with the bacteria. In addition, the structural changes of the PP before and after strain infusion were assessed through infrared spectroscopy.

## 2. Materials and Methods 

### 2.1. Polypropylene Microplastics

The PP microplastics were attained by means of cutting and grating the commercial PP plastic materials obtained from plastic-producing industries. The plastic samples were sieved through a sieve stacker (sieve size ranging from 4 mm to 250 μm), and the particles with a size of 1 mm were collected. The microplastic samples were then washed with ethanol and oven-dried (60 °C) overnight or until sample dryness.

### 2.2. Microorganism and Microbial Inoculum Preparation

The bacterial strains used in this study were isolated from the samples of fellfield soils collected from Victoria Land (Ross Sea sector Antarctica) [30]. These pristine soil were initially enriched with diesel fuel as the sole carbon source prior to the selection of appropriate bacterial strains. The strains were identified through 16S rRNA sequence analysis and designated as *Pseudomonas* sp. ADL15 (GenBank accession No. KX812776) and *Rhodococcus* sp. ADL36 (GenBank accession No. KX812777) [31]. The strains were previously stored at −80 °C in 80% (*v/v*) glycerol solution. The bacteria were revived on nutrient agar (NA) before inoculation into the nutrient broth (NB) and left to grow at 10 °C for 3 days in an incubator shaker until the log phase.

### 2.3. Assay for PP Microplastic Utilisation

Prior to the experiment, the PP microplastic samples were disinfected with ethanol (30 min soaking) and air-dried in the laminar airflow chamber. For the assessment of growth and biodegradation, BH broth medium consisting of MgSO_4_ (0.2 g/L), CaCl_2_ (0.02 g/L), K_2_HPO_4_ (1.0 g/L), KH_2_PO_4_ (1.0 g/L), NH_4_NO_3_ (1.0 g/L), and FeCl_3_ (0.05) was used throughout the entire experiment. No carbon source was supplemented to the medium to eliminate the strains’ growth-reliance towards the supplementary carbon. The pH of the medium was adjusted to 7.0 ± 0.2 at 25 °C before sterilisation. The assay was evaluated by inoculating 10% of the bacterial culture (OD_600_ = 1.0) into liquid culture (45 mL of BH media with 0.1 g of PP microplastics in 250 mL conical flask). The flasks were then shaken in an incubator shaker (10 °C, 150 rpm). The uninoculated BH medium with PP microplastics was used as the negative control. The optical density (OD_600_), pH, and microbial counts were measured and calculated every 10 days for each flask for 40 days. All experiments were performed in triplicate.

### 2.4. Determination of Weight Loss of Residual PP Microplastics

The remaining PP microplastics were retrieved from the medium through filtration, sequentially washed with 70% ethanol solution for maximum removal of cells and debris, and oven-dried (60 °C) overnight or until sample dryness. The final weight of the polymer was measured using the analytical balance (HRB Series Tree Balance) with a sensitivity of 0.001 g. The initial weights of the polymer were also measured before incubation. The weight loss percentage of PP microplastics was determined using Equation (1):% weight loss = ((*W*_0_ − *W*)/*W*_0_) × 100,(1)
where *W*_0_ is the initial weight of the PP microplastics (g) and *W* is the final weight of the PP microplastic (g).

### 2.5. Determination of Polymer Reduction Rate and Half-Life of Residual PP Microplastics

The initial and final weights obtained were used to calculate the rate constant of PP microplastic reduction using the first-order kinetic model with specific intervals of 10 days [28]. The polymer reduction rate was calculated as follows (Equation (2)):*K* = −1/*t* (ln (*W*/*W*_0_)),(2)
where *K* is the first-order rate constant for PP microplastic uptake per day, *t* is time in days, *W* is the final weight of PP microplastics (g), and W_0_ is the initial weight of PP microplastics (g). 

The respective formula and model was utilised and employed as it gives a constant fraction of PP microplastic removal per unit time. The half-life of a first-order reaction is independent of the original concentration of the substrate. Employing the acquired value of microplastic removal rate constant, *K*, the half-life was calculated as follows (Equation (3)):*t*_1/2_ = ln (2)/*K*,(3)

### 2.6. Fourier Transform Infrared (FTIR) Analysis of PP Microplastics

The infrared spectroscopic analyses were performed on Spectrum 100 1FTIR spectrometer (PerkinElmer, Waltham, MA, USA) with the 5 mg polymer sample dispersed in pellets of KBr to elucidate the types of chemical bonds (functional groups). The measurements were conducted within the 4000–400 cm^−1^ wave number. The normalised intensity of the functional groups was determined based on the transmittance (%). Analyses were conducted on both PP microplastic samples incubated with ADL15 and ADL36 and on the uninoculated control PP microplastic samples.

### 2.7. Statistical Analysis

Due to the presence of two independent groups (ADL15 and ADL36) and small sample size, the statistical analysis of data was carried out using the nonparametric Mann–Whitney *U* test (SPSS software, version 25.0) to evaluate the weight loss of polypropyelene (PP) microplastics. 

## 3. Results and Discussion

### 3.1. Growth of Bacteria on PP-Supplemented Medium and Estimation of Weight Loss, Reduction Rate, and Half-Life of PP Microplastics

The pursuit of microorganisms from particular environments such as the Antarctic is due to the unique characteristics that the microbes may possess, such as tolerance towards the cold temperature and high survivability in an environment lacking in nutrients and water availability [32]. Previous studies revealed that *Pseudomonas* and *Rhodococcus* species are widely available in either pristine or contaminated Antarctic soils [31,33]. Both strains used in this study were previously isolated from pristine fellfield soils collected from Victoria Island, Antarctica. The strains’ remarkable features include the utilisation of diesel [31] and the production of biosurfactant [34,35]. Strain ADL36 also able to utilise waste palm oil as the sole carbon source [36]. Owing to these attributes, the assessment of the ability of these strains to utilise the highly hydrophobic PP microplastics as their energy source appears to be promising.

In this experiment, the Antarctic strains were cultured and retained on the NA plates at 10 °C for 3 days. The strains were screened for the potential of PP microplastics utilisation as their carbon source. Positive growth was observed for both strains on the basis of optical density (OD_600_) and viable plate counts (Figure 1a,b), which coincided with one another. From the figures, it can be inferred that ADL15 demonstrates more tolerance towards the exposure to PP microplastics compared to ADL36. This can be observed when comparing the overall pattern of the growth curve of ADL15 and ADL36 in 40 days. The former strain did not show the highest growth compared to ADL36. Yet, the small growth of ADL15 could degrade a higher percentage of PP microplastic (*Mdn* = 17, n = 3) than ADL36 (*Mdn* = 7, n = 3) (*U* = 0.000, *p* = 0.043, *r* = 0.825), as shown in Figure 1c. The growth of these bacterial isolates might differ due to their distinctive metabolic rates, genetic plasticity, polymer preferences, and mode of substrate colonisation [37]. The higher tolerance of ADL15 towards the exposure of PP could lie within the genus itself. In a review by Wilkes and Aristilde [38], the genus *Pseudomonas* is highlighted as among the highly sought bioremediation agents for hydrophobic polymers due to their unique cell-surface attachment, numerous catalytic enzymes, and extensive metabolic pathways specialised for plastic polymers. The biosurfactant-synthesising ability of ADL15 could also be a significant factor that contributes to a more hydrophilic PP form [34]. Besides facilitating the attachment of bacterial cells to the polymer surface, a relatively hydrophilic PP is more susceptible to microbial attack and subsequent degradation.

Though ADL15 appears to be a better candidate to utilise PP, the significantly higher growth of ADL36 should not be overlooked. ADL36 seems to be a good PP-colonising bacteria as the strain could grow in bulk with the little amount of PP in the medium. The slow but gradual increase of growth of ADL36 supports the theory of the genus *Rhodococcus* as the *k*-strategist, where they reproduce at a slower rate while maintaining a stable ambience (using less PP for growth) where new progeny is formed with a greater chance of survival and persistence [39].

Compared to the growth of bacterial strains from Auta et al. [28], the Antarctic strains exhibit lower optical density and viable plate count, indicating a slower growth of strains than the previous report within the same period. This might be due to the experimental setup at a lower temperature (10 °C) where either growth or degradation of synthetic compounds are expected to be at a slower rate. However, the aptitude to utilise PP as the growth supply at a low temperature presents a remarkable quality for both strains.

Biodegradation of microplastics is a result of the utilisation of the plastic substrate as the microbes’ carbon source to support their growth. Previously, the extent of biodegradation of polymers was examined through morphological changes, weight/mass loss, and the decrease of tensile strength and molecular weight [29]. In this study, the utilisation of untreated PP microplastics by the Antarctic strains was determined quantitatively via the percentage of weight loss of the polymers after 40 days of incubation. The results are represented in Figure 1c and summarised in Table 1. The positive weight reduction implies that specific catabolic enzymes were synthesised by the isolates and attacked the untreated PP microplastic, leading to the polymer’s partial degradation and subsequent reduction. As mentioned earlier, these isolates may also possess a particular signalling and catabolic pathway that promotes the cell adherence to the polymers and their subsequent absorption/desorption and degradation [37]. Lack of degradation in the control flask was indicated by the constant weight of PP microplastics throughout the experiment.

Previous reports have observed that the ability to utilise PP micropastics was dominantly shown by several members of the class Bacilli (Table 2). Degradation of PP microplastics by *Bacillus gottheilii* recorded a weight loss of 3.6% after 40 days [40]. The bacterial strains *Bacillus cereus* and *Sporosarcina globispora* that were isolated from the Malaysian mangrove ecosystems were able to degrade PP microplastics with the approximate weight loss of 12% and 11% in 40 days, respectively, which was the highest weight loss of PP microplastics recorded by bacteria at that point [41]. Another study by Auta et al. [28] also reported a weight loss of 4.0% and 6.4% of PP microplastics by using a locally-isolated *Bacillus* sp. strain 27 and *Rhodococcus* sp. strain 36, respectively. With regards to the genus *Rhodococcus*, strain ADL36 used in this study showed a higher percentage of weight loss (7.3%) than the strain 36 reported by the former study, indicating a higher affinity for PP microplastics. Fontanella et al. [29] previously reported that *Rhodococcus rhodochrous* ATCC 29672 is capable of degrading PP films containing pro-oxidant additives based on metal ions to a certain degree in 6 months. *Pseudomonas* sp. ADL15 recorded an approximate 17.33% weight loss after incubation with neat PP after 40 days. Currently, the percentage of PP weight loss recorded by ADL15 within 40 days is the highest for any single microorganism, with or without pretreatment. Previously, biodegradation of thermally and UV-treated PP by the genus *Pseudomonas* reported by Arkatkar et al. [27] achieved the percentage of weight loss between 0.6% and 1.5%. Aravinthan et al. [42] also reported on the biodegradation of pr-treated PP using a *Pseudomonas* and *Bacillus* mixed consortium where a thermogravimetric (TG) and a gravimetric weight loss percentages of 22.7% and 1.95 ± 0.18%, respectively, were achieved.

The determination of the removal rate constant (*K*) of the PP microplastics per day by the Antarctic bacterial isolates is via the first-order kinetics and the half-life, *t*_1/2_. The half-life was calculated to estimate the time necessary for half of the PP microplastics to be reduced. Based on Table 1, ADL15 recorded a higher removal rate of 0.0047 day^−1^ and a shorter half-life of approximately 147 days, compared to strain ADL36. ADL36 recorded a lower average uptake rate of 0.0018 day^−1^ and a longer average half-life of approximately 385 days to reduce half of the 0.1g of PP microplastics. This study recorded the fastest removal rate constant, *K*, and half-life, *t*_1/2_, for the degradation of PP microplastics by bacterial strains. However, the high removal rate constant and short half-life in our study are believed to be affected by the smaller size of PP microplastics used (0.1 g) when compared to previous studies [28,40].

### 3.2. Changes in the pH of the Bushnell Haas (BH) Media

Biodegradation of plastics is primarily influenced by environmental factors such as pH, as pH has a profound influence on the survival of microbes and the success of the hydrolytic cleaving of the “microwastes” [14,44]. Figure 1d shows the pH changes of the BH media infused with PP microplastic during the 40 days of incubation with Antarctic bacterial strains. The changes in the pH for the liquid medium inoculated by the two strains increased from neutral to slightly alkaline along the incubation period, although there were bacterial growth inconsistencies. Although there is no definite comprehension of the pH increment during the course of polymer degradation, the continuous increase of pH towards the alkaline state is believed to happen because of the production and accumulation of oligomers and intermediate products via the action of bacterial exoenzymes [45]. From the 0th to 40th day, the pH progressed from 7.12 to 8.17 for ADL15 and 7.17 to 7.94 for ADL36. The optimum growth of ADL36 (OD_600_ = 0.776, 13.39 log CFU/mL) was achieved on the 30th day when the pH was 7.89, whereas ADL15 (OD_600_ = 0.481, 13.22 log CFU/mL) achieved optimal growth on the 40th day when the pH reached 8.17. While a further increase in the pH (7.89 to 7.94) in the last 10 incubation days led to the decrease in the growth of ADL36, the growth reduction appears to be insignificant (*p* > 0.05) (Figure 1a). As for ADL15, since the optimal pH growth was on the 40th day, further incubation of the bacteria might be required to observe the trend between the bacterial growth and pH of media. 

### 3.3. Changes in the PP Microplastic Structure

The changes in the PP microplastic structure were determined using FTIR spectroscopy at a frequency range of 4000–400 cm^−1^. Figure 2a shows the FTIR spectrum of control (disinfected PP microplastic) incubated without the presence of bacterial strains for 40 days in BH media. Several significant peaks were observed in the infrared spectrum of the disinfected PP microplastics. The occurrences of C-H alkyl stretch were observed via the presence of collective sharp absorption peaks at 2929, 2917, 2867, and 2838 cm^−1^. Two minor but sharp bands at 1456 and 1376 cm^−1^ represents the alkane bends of methylene (CH_2_) and methyl (CH_3_) groups, respectively. The absorption peak at 1166 cm^−1^ corresponds to the C-O phenolic bands. The absorption peaks between 800–900 cm^−1^ signify the C-H alkyl bend of the raw PP.

With the inoculation of the Antarctic strains into the PP-infused BH media, minor changes of the PP microplastic structure were observed through the infrared spectra. Firstly, there is a shift of the whole spectrum along the y-axis (transmittance) for the ADL15-treated PP, although it does not constitute any significant changes to the polymer structure (overlapping of these spectra showed similar pattern) (Figure 2b). No shift was detected in the infrared spectrum of ADL36-treated PP (Figure 2c). Secondly, both ADL15- and ADL36-treated PP microplastic samples showed significant degradation or bond-breakage of the polypropylene molecules. This can be observed through the increase of transmittance value at several absorption peaks (2839–2951, 1456, 1376, and 1167 cm^−1^) found in the sample. High transmittance at a frequency signifies the presence of few bonds that absorb the light in the sample, while the low transmittance value denotes a greater number of bonds that possess vibrational energies that correspond to the incident light. The increase of the transmittance at the observed peaks indicates the utilisation of the major alkyl group (C-H) by both strains as their energy source. Finally, the presence of a slight “tongue-like” peak after the 3310 cm^−1^ region in the infrared spectra in Figure 2b,c might be attributed to the hydroxy group (-OH). The unexpected appearances of the hydroxy group are somewhat analogous in the infrared spectra for UV-treated PP [28]. This suggests that the infusion of PP with the Antarctic bacteria may be comparable to the effect of UV irradiation to a certain extent.

## 4. Conclusions

The present work provides the first report on the biodegradation of microplastics by Antarctic soil bacteria and highlights the potential of *Pseudomonas* sp. ADL15 and *Rhodococcus* sp. ADL36 for biodegradation of polypropylene (PP) microplastics. Both isolates demonstrated positive growth in the medium containing PP microplastics. Among the two strains, ADL15 was characterised by a more efficient utilisation, higher relative weight loss, faster removal rate constant, and a shorter half-life to biodegrade PP microplastics compared to ADL36. The isolates’ concurrent aptitude to grow at low temperatures and degrade a certain degree of raw PP offers new information to the area of study (microplastic biodeterioration), possibly creating a new direction for the research on the whole spectrum. 

The ability of the Antarctic strains to deteriorate neat PP microplastics mirrors the actual PP biodeterioration in the soil environment, which has limited exposure to photo-oxidation. The use of commercial PP products from the plastic-producing industries in the present study also mimics the real improper waste disposal of standard/commercial PP products in the environment. However, additional approaches such as thermal and UV pretreatment of the polymer should also be observed to understand any acceleration process in biodegradation. Following the paths of preceding research, structural and morphological study using scanning electron microscopy (SEM) should be employed to fully comprehend the authentic modifications or transformations of the PP. Following the big data explosion of genomics and transcriptomics, the respective enzymes and pathways and the corresponding mechanisms involved in the biodegradation of microplastics should also be explored further in detail. Then again, the lack of a coherent working scheme for the biodeterioration study of polypropylene that can drive to conclusive postulations has limited our ability to create a biochemically based comprehension of the mechanism and processes involved in the PP microplastic degradation.

## Figures and Tables

**Figure 1 polymers-12-02616-f001:**
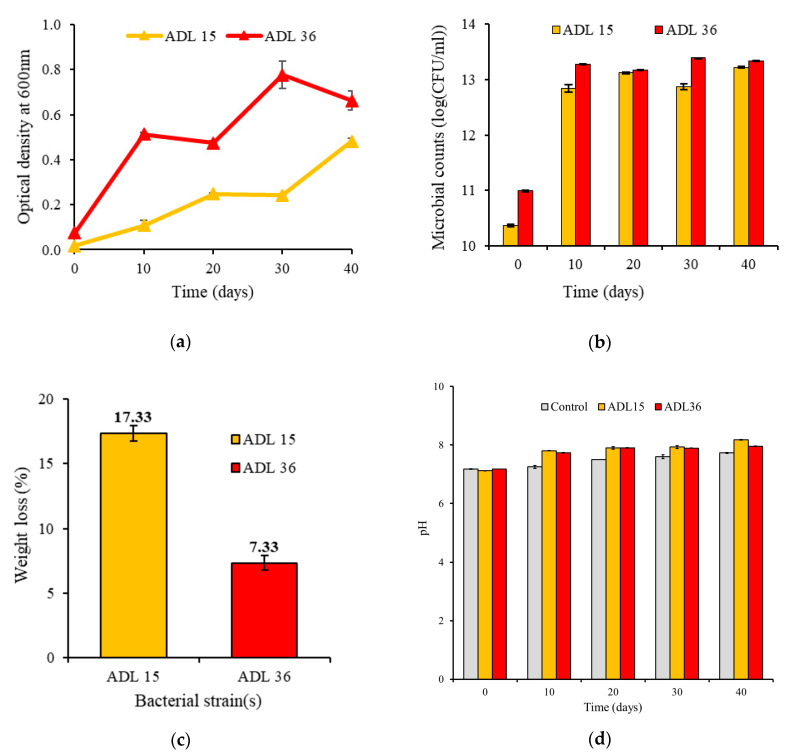
Changes in cultures of the Antarctic bacteria *Pseudomonas* sp. ADL15 and *Rhodococcus* sp. ADL36 during 40 days of incubation in Bushnell Haas (BH) media infused with polypropylene (PP) microplastics: (**a**) optical density, (**b**) microbial counts, (**c**) relative weight loss of PP microplastics, and (**d**) pH of the media. Data represent mean ± SD, n = 3.

**Figure 2 polymers-12-02616-f002:**
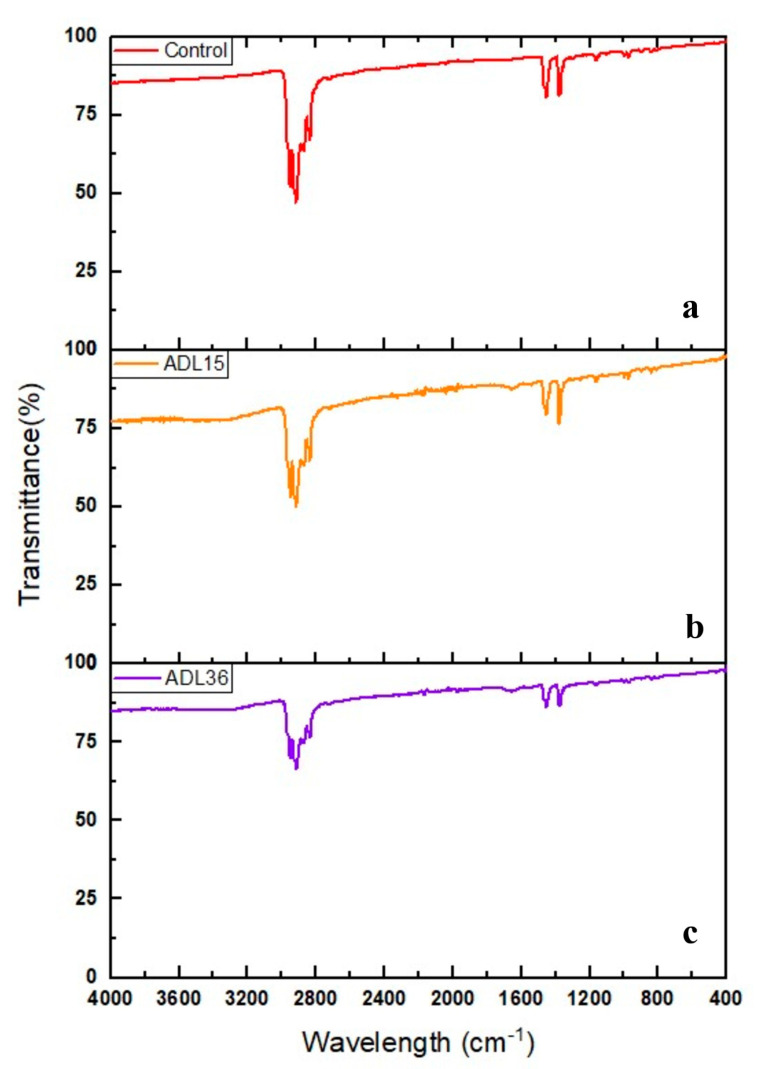
The Fourier transform infrared (FTIR) spectrum of polypropylene (PP) microplastics: (**a**) before incubation with bacteria, (**b**) after 40 days of incubation with *Pseudomonas* sp. ADL15, and (**c**) after 40 days of incubation with *Rhodococcus* sp. ADL36.

**Table 1 polymers-12-02616-t001:** Summary statistics (Mann-Whitney *U* test) of polypropylene (PP) microplastic weight loss in cultures of the Antarctic bacteria *Pseudomonas* sp. ADL15 and *Rhodococcus* sp. ADL36.

Strain	Initial Weight (g)	Final Weight (g)	Weight Loss (%)	*U*	*Z*	*p*	*r*	Removal Rate Constant, *K* (Day^−1^)	Half-Life (Days)
Control	0.100	0.100	0.0	not determined				0	∞
ADL15	0.100	0.083 ± 0.0006	17.3	0.000	−2.023	0.043	0.825	0.0047	147
ADL36	0.100	0.093 ± 0.0006	7.3					0.0018	385

**Table 2 polymers-12-02616-t002:** The polypropylene-degrading bacterial strains and consortia.

Strain (s)	Origin	PP Treatment (s)	Period	SM ^1^	T ^2^ (°C)	Degradation (%)	Reference (s)
Mixed soil culture (consisting of *Bacillus flexus* strain of [27])	Palikarne dumping ground, Chennai, India	PP films (0.05 mm) Thermal pretreatment 80 °C for 10 days (PP-TT) Untreated (PP-UT)	1 year	n.d ^3^	30–37	0.43 (PP-UT) 10.7 (PP-TT)	[21]
*B. flexus*		PP films (0.05 mm) Chemical pretreatment * Thermal and photo-pretreatment 100 °C for 8 days (PP-TT) or short ultraviolet (UV) (PP-SUV) at 225 nm for 6 days		BF ^5^, BS ^6^	35–37	>2.5 (PP-SUV) >0.7 (PP-TT)	[27]
*Bacillus subtilis*						>1.5 (PP-SUV) >0.6 (PP-TT)	
*Pseudomonas azotoformans* ^4^						>0.5 (PP-SUV) >0.5 (PP-TT)	
*Pseudomonas stutzeri*				BS		>1.25 (PP-SUV) >0.6 (PP-TT)	
Mixed consortia		PP films (0.05 mm) Thermal and photo-pretreatment 100 °C for 8 days (PP-TT) or short ultraviolet (UV) (PP-SUV) at 225 nm for 6 days		BF, BS	28 ± 2	1.95 ± 0.18 (PP-UV) 1.12 ± 0.04 (PP-TT) 22.7 (PP-UV/TG ^7^) 16.5 (PP-TT/TG)	[42]
*B. flexus* + *P. azotoformans* (B1)							
*B. flexus* + *B. subtilis* (B2)						1.48 ± 0.11 (PP-UV) 1.22 ± 0.22 (PP-TT) 17.6 (PP-UV/TG) 15.02 (PP-TT/TG)	
*Stenotrophomonas panacihumi* PA3-2	Storage yard, Gyeonggi-do, Korea	Low molecular weight PP (LMWPP-1 and -2) No pretreatment	90 days	n.d	37	20.3 ± 1.39 (LMWPP-1) 16.6 ± 1.70 (LMWPP-2)	[20]
*Bacillus cereus*	Mangrove ecosystem, Peninsular Malaysia	Isotactic PP granules	40 days	n.d	RT ^8^	12	[41]
*Sporosarcina globispora*						11	
*Bacillus gottheili*		Isotactic PP granules Ultraviolet treatment for 25 days				3.6	[40]
*Bacillus* sp. 27						4.0	[28]
*Rhodococcus* sp. 36						6.4	
*Rhodococcus rhodochrous* ATCC 29672	ATCC ^9^	Extensive pretreatment	180 days	n.d	27	n.d	[29]
Thermophilic consortia *Aneurinibacillus* sp. ISI *Brevibacillus* sp. IS3, ISA, ISC	Districts in Karnataka state, India	PP strips (0.15 mm) (PPS) PP pellets (2.5 mm) (PPP)	140 days	BF	50 ^10^	56.3 ± 2 (PPS) 44.2 ± 3 (PPP)	[43]
*Pseudomonas* sp. ADL15	Soils from Victoria Island, Antarctica	Grated PP (1 mm) No pretreatment	40 days	n.d	10	17.3	This study
*Rhodococcus* sp. ADL36						7.3	

^1^ Secondary metabolite; ^2^ temperature; ^3^ not determined; ^4^ nonisolated strain (provided by National Environmental Engineering Research Institute (NEERI)); ^5^ biofilm; ^6^ biosurfactant; ^7^ thermogravimetric; ^8^ room temperature; ^9^ American Type Culture Collection; ^10^ other temperatures were also investigated (5, 25, 47, 45, and 55 °C); * no weight loss recorded.

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
