# Peer review of "Biodeterioration of Untreated Polypropylene Microplastic Particles by Antarctic Bacteria"

_polymers, 2020, doi:10.3390/polym12112616_

Round 1

Reviewer 1 Report

Overview:

The manuscript reviewed investigated the growth and biodegradation potential of Antarctic soil bacteria (Pseudomonas sp. ADL14 and Rhodococcus sp. ADL36). The study was done over 40 days on polypropylene in a Bushnell-Haas medium. Biodegradation was assessed by percentage weight loss of PP.

The research conducted was novel and of significance for scientists involved microplastics (MPs) research. The written presentation was good with a clear flow of ideas. Some editing is suggested below that the authors may consider. The statistical analysis and information regarding control groups need to be addressed and I recommend that the editor consider these suggestions before making a decision to accept the manuscript for publication.

General comments:

Check manuscript and edit grammar; provide more information regarding statistical analysis and control groups.

Specific comments / suggestions:

Line(L)17-18: edit: microplastic pollution in the soil environment is generally little understood and often overlooked;

L34-36: edit: Microplastics are considered as the minute plastic particle with a size of < 5 mm which caused severe environmental pollution, due to their exponential increase in large scale manufacturing, mass production, and vast utilisation in the world;

L40: edit: their environmental aftermath;

L42: edit: have only been commenced for these past few years;

L49-50: edit: they initially settle on the surface before penetrates the subsoils;

L50-51: edit: The disintegrated microplastics then would be further penetrated deeper soil layers by the act of the soil occupants such as collembolans, earthworms, and plants;

L54: edit: in a small degree;

L55-56: edit: The limited exposure of photo-oxidation towards the soil-incorporated microplastics than those on the surface only aggravates the situation;

L58-60: edit: Certain microorganisms that are found on the deeper soil colonises the plastic polymers where biodegradation was initiated through the adherence of microorganisms to the plastic surfaces – facilitated via the formation of biofilm;

L60-63: suggest inserting a reference;

L65-66: edit and provide clarity: The characteristics of these soils are strictly associated with the uncontrolled biotic and abiotic factors which highly correlated to the geographical aspects;

L68: suggestion: replace ‘climate’ with environments / environmental conditions?

L74: suggestion: replace ‘divulges’ with indicates?

L76-78: edit: Application-wise, the utilisation of polypropylene (PP) is widely comprised from the single-use disposables to the long-lasting durables due to its excellent features – exceptional mechanical properties, simple manufacturing, and reasonably priced;

L82-84: suggest inserting a reference;

L90-91: edit: Microorganisms that can grow under extreme temperature and capable to use neat microplastic as their sole energy source have never been reported yet.;

L92: suggestion:  make reference to the bacteria?

L96: edit: were assessed through the infrared spectroscopy;

L99-100: suggestion: indicate how the PP were cut and grated;

L101: what other sieve sizes were included? these are not referred to in the results - relevance?

L103: edit: until sample dryness;

L112: provide more information/edit: until the log phase;

L119: suggestion: explain how the pH was adjusted;

L142: edit: was utilised and employed;

L153-154: more information regarding the statistical analysis in required. Did the data meet the assumptions for ANOVA analysis (normality and variance)? Which parameters were analysed? From the results it appears that statistics was done on 2 groups, ADL 15 and ADL 36 (eg Fig 1). If 2 groups were statistically analysed, why not use a T-test? Given that there were only 3 replicates, the use of parametric analysis is questionable and must be addressed before the results can be accepted.

Section 3: Results: reference to statistical analysis is not presented in the result section of the manuscript (information in Table 1 is inadequate) and must be addressed.

L158-160: edit: The pursuit of microorganisms from particular environments such as the Antarctic is due to the unique characteristics that the microbes may possess such as tolerance towards the cold temperature and high survivability on the environment with lack of nutrients and water availability;

L165-166: edit: Owed to these attributes;

L171: Fig 1a and 1b: were there any control groups? If so these need to be included in the figures. If not, it severely weakens the value of the results without being able to compare to a control group.

Fig 1a-d: what do error bars indicate? SD/SEM/CI?

L174-176: were the differences in weight loss significant between the 2 groups?

L203-204: edit: Biodegradation of microplastics is resulted by the utilisation of the plastic substrate as the microbes’ carbon source to support their growth;

L204-206: edit: Previously, the extent of biodegradation of polymers is examined through morphological changes, weight/mass loss, and the decrease of tensile strength and molecular weight;

L208/L213-214: Fig 1c: was there a control group for this? Could handling and washing with 70% ethanol have affected weight loss. Any other factors?

Table 1: what do the F and r^2 values indicate? What was the variance of average weight loss?

L220: clarify reference to ‘utilise’;

L219-230: suggestion: include a table where comparisons are made to results from other investigations.

L253: Fig 1d: was there a control group?

L264-265: caution is advised regarding use of the terms ‘the growth reduction appears to be insignificant’. Why not use a statistical test to determine significance?

L280: suggestion: reference to ‘transmittance’ needs to be explain in the M&M section where the FTIR settings are indicated;

Fig 2: suggestion: the 3 graphs can be merged into 1 graph – separate y-axis lines with the same x-axis;

L319: suggestion: reconsider reference to ‘the big data explosion’;

L321-324: edit: Then again, the lack of a coherent working scheme for the biodeterioration study of polypropylene that can drive to conclusive postulations has limited our ability to create a biochemically-based comprehension of the mechanism and processes involved in the PP microplastics degradation.

Reviewer 2 Report

It is an interesting research.

The other temperature should also be investigated, maybe a higher degradation efficiency could be got, also, the lower T should be tested (such as the environmental T of antarctic)

The degradation products should be investigated. If the products was toxic or not?

The details of degradation experiments should be given.

Reviewer 3 Report

Dear Editor,

the manuscript  presented by Habib et al. studies  the ability of two bacteria strains isolated from Antartica  to degrade polypropilene plastic. The paper was very easy to read. However, it miss of microscopical analyses that should evidence the presence of biofilm and eventual biodeterioration.

The Title is not appropriate:

Biodeterioration should be studied with more analyses including microscopy and biofilm formation.

So in the present form the title should be modified as "Biodeteriorative of polypropylene microplastic particles"abillities  of two antartic bacteria

In addition I suggest to the authors to add more information about the characteristics of  two bacterial strains that were identified only  at genus level.

Round 2

Reviewer 1 Report

It is a pity the authors did not consider any of the suggested edits, as this decreases the quality of the submission. My recommendation is still for the authors to make the changes, but will leave this for the Editor to decide.

Comment 25: the statistical analysis is fundamentally flawed in its current form in my opinion. In order to do a T-test, the data must meet the assumptions that the variances were similar and that the data was normally distributed. Without providing information regarding this, the statistical analysis becomes questionable. Evidence of these assumptions been met are not provided. If the data did not meet these assumptions, either try analyses using a GLLM or a non-parametric analysis such as a Mann-Whitney test (for the 2 groups). 

Comment 26: It is advised to describe results presented in the Table in the results section. 

Comment 29: The authors should then show the results / data for the control group at the start and end of the experiment. It is necessary as it shows the reader what to compare the experimental data / results with. Without it, the results are not comparable in my opinion.

Comment 31: The significant differences should be described in the text, not only the Table.

Comment 34: The authors should then show the results / data for the control group at the start and end of the experiment. It is necessary as it shows the reader what to compare the experimental data / results with. without it, the results are meaningless in my opinion.

Comment 35: the F value refers to the Levenes test as per the heading in Table 1. It does not refer to the "F statistics which correlated to the significance (Sig.)" as per the authors response.

What the is that F value referring to? A Levenes test is a test for variance of data.

Comment 38: The authors should then show the results / data for the control group at the start and end of the experiment. It is necessary as it shows the reader what to compare the experimental data / results with. without it, the results are meaningless in my opinion.

Round 3

Reviewer 1 Report

The authors have addressed most of the suggestions made. If the editor is happy with the submission and changes made, I recommend that the manuscript be accepted for publication.